# Hepatokines and MASLD: The GLP1-Ras-FGF21-Fetuin-A Crosstalk as a Therapeutic Target

**DOI:** 10.3390/ijms251910795

**Published:** 2024-10-08

**Authors:** Ilaria Milani, Michela Codini, Gloria Guarisco, Marianna Chinucci, Chiara Gaita, Frida Leonetti, Danila Capoccia

**Affiliations:** 1Department of Medico-Surgical Sciences and Biotechnologies, Faculty of Pharmacy and Medicine, University of Rome La Sapienza, 04100 Latina, Italy; ilaria.milani@uniroma1.it (I.M.); gloria.guarisco@uniroma1.it (G.G.); chinuccimarianna@gmail.com (M.C.); chiara.gt17@gmail.com (C.G.); frida.leonetti@uniroma1.it (F.L.); 2Department of Pharmaceutical Sciences, University of Perugia, Via Fabretti 48, 06123 Perugia, Italy; michela.codini@unipg.it

**Keywords:** steatosis, metabolic dysfunction, hepatokines, liver disease

## Abstract

The introduction of the term “Metabolic Steatotic Liver Disease” (MASLD) underscores the critical role of metabolic dysfunction in the development and progression of chronic liver disease and emphasizes the need for strategies that address both liver disease and its metabolic comorbidities. In recent years, a liver-focused perspective has revealed that altered endocrine function of the fatty liver is a key contributor to the metabolic dysregulation observed in MASLD. Due to its secretory capacity, the liver’s increased production of proteins known as “hepatokines” has been linked to the development of insulin resistance, explaining why MASLD often precedes dysfunction in other organs and ultimately contributes to systemic metabolic disease. Among these hepatokines, fibroblast growth factor 21 (FGF21) and fetuin-A play central roles in regulating the metabolic abnormalities associated with MASLD, explaining why their dysregulated secretion in response to metabolic stress has been implicated in the metabolic abnormalities of MASLD. This review postulates why their modulation by GLP1-Ras may mediate the beneficial metabolic effects of these drugs, which have increased attention to their emerging role as pharmacotherapy for MASLD. By discussing the crosstalk between GLP1-Ras-FGF21-fetuin-A, this review hypothesizes that the possible modulation of fetuin-A by the novel GLP1-FGF21 dual agonist pharmacotherapy may contribute to the management of metabolic and liver diseases. Although research is needed to go into the details of this crosstalk, this topic may help researchers explore the mechanisms by which this type of pharmacotherapy may manage the metabolic dysfunction of MASLD.

## 1. Introduction

Metabolically associated steatotic liver disease (MASLD), formerly known as non-alcoholic fatty liver disease (NAFLD), is the most common chronic liver disease worldwide. It results from the interaction of multiple cardiometabolic and environmental risk factors [1] and encompasses a broad spectrum of pathological conditions, including metabolic steatohepatitis associated with liver dysfunction (MASH), which can progress to cirrhosis [2]. Given the strong association between steatotic liver disease and complex metabolic abnormalities, it has taken several years to determine which terminology is more appropriate. The inclusion of metabolic dysfunction in the MASLD diagnostic criteria [3] allows both the identification of at-risk individuals and the development of strategies that target liver disease and metabolic comorbidities [4].

Changes in the pattern of hepatokine secretion due to liver dysfunction associated with steatotic liver disease appear to contribute directly or indirectly to impaired metabolic homeostasis. In this context, hepatokines may be recognized as biomarkers of ectopic fat accumulation in the liver and progression of steatotic liver disease [5]. Their probable diagnostic and prognostic roles suggest that these molecules may be considered as targets for the prevention and treatment of diseases associated with metabolic dysfunction, including MASLD.

This review article summarizes the complex and multifaceted relationships between MASLD and these hepatokines, with a particular focus on two of the most widely studied hepatokines in humans and animals [6]. In addition, it has been hypothesized that the possible role of glucagon-like receptor agonists (GLP1-Ras) in the modulation and interaction of these two hepatokines could provide an explanation of the therapeutic potential for the treatment of MASLD with these drugs.

## 2. MASLD: Epidemiology and Diagnosis

Globally, the introduction of new diagnostic criteria for MAFLD/MASLD has led to an increasing trend in the prevalence of the disease, estimated at 50.4% between 2006 and 2019, with the highest prevalence in Europe, Asia, and North America [4]. Specifically, the prevalence of MASLD will increase from 22% to 37% in the adult population between 1991 and 2029 [7], ranging from 13.5% in Africa to 31.8% in the Middle East, and increasing with age from 30.7 to 76.2 years.

Given the multifactorial nature of MASLD, it is not surprising that MASLD is more common in the presence of cardiometabolic risk factors such as type 2 diabetes (T2D), obesity, hypertension, dyslipidemia, and metabolic syndrome [8], with a growing parallel prevalence [7]. MASLD has been estimated to be present in 55.5% of patients with T2D [8], 80% in those with obesity, and 72% in those with dyslipidemia [9], increasing the number of cases of cirrhosis, HCC, hepatic decompensation, and liver-related mortality [4].

If the diagnosis of NAFLD is based on the exclusion of significant alcohol use and other etiologies of fatty liver disease as unique criteria [7], the diagnosis of MASLD is based on positive criteria, including evidence of hepatic steatosis by imaging or biopsy, along with at least one of five cardiometabolic abnormalities, such as the presence of overweight or obesity, impaired glucose regulation or T2D, hypertension, elevated plasma triglycerides, or decreased high-density lipoprotein cholesterol (HDL-c) [7]. Because Homeostatic Model Assessment for Insulin Resistance (HOMA-IR) and high-sensitive C-reactive protein (hs-CRP) levels are not measured in all clinical settings, the MASLD criteria are more intuitive than the MAFLD criteria [10].

In addition, although MASLD almost overlaps with the NAFLD population, the inclusion of cardiometabolic risk factors should allow the identification of new phenotypes as the field evolves [1], with a slightly higher mortality risk in MASLD than in NAFLD [11]. In a population-based cohort study, the risk of death from any cause increased with the severity of MASLD, with an estimated absolute excess risk of death of 10.7% for patients with simple steatosis, 18.5% for MASH without fibrosis, 25.6% for fibrosis without cirrhosis, and 49.4% for cirrhosis compared with controls [8].

Therefore, the new nomenclature of MASLD not only emphasizes the critical pathogenetic role of systemic metabolic dysfunction in the process of chronic liver steatosis but also raises clinicians’ awareness of the concomitant metabolic dysfunction in patients with MASLD [12].

## 3. MASLD: Risk Factors and Pathogenetic Mechanism

Multiple factors are involved in the development of steatosis, the overlap of which has led to the identification of several clinical pathophysiological subphenotypes with different adverse outcomes. Indeed, MASLD is strongly associated with T2D, dyslipidemia, and insulin resistance, an overlap that poses a challenge to understanding the pathogenesis of this disease [9] and led to its identification as a hepatic manifestation of the metabolic syndrome [12].

These factors were divided into three categories (Figure 1), such as individual characteristics (age, sex, ethnicity, diet and physical activity, alcohol consumption, and microbiota composition), metabolic factors (body mass index (BMI) and visceral adiposity, diabetes, hypertension, and dyslipidemia), and genetic/epigenetic factors (SNP in patatin-like phospholipase domain-containing 3 (PNPLA3) p.I148M and transmembrane 6 superfamily member 2 (TM6SF2) p.E167K, which have been shown to increase MASLD progression along with metabolic health and individual characteristics). In any case, the number and severity of metabolic abnormalities (obesity, T2D, and dyslipidemia) increase the progression of MASLD from simple steatosis to MASH with or without fibrosis [9].

A meta-analysis reported a prevalence of MASLD in the overweight and obese population of 69.99% and 75.27%, respectively [13]. Abdominal adipose tissue dysfunction is also recognized as a major risk factor in lean individuals with MASLD [9], with an estimated prevalence of 10.2%–15.7% in the normal weight population according to another meta-analysis [14].

Given the complexity of the pathophysiology of MASLD, the heterogeneity of MASLD in the clinical context and its progression was initially attributed to the classic “two-hit theory”, with excessive hepatic lipid deposition recognized as the first hit, followed by activation of inflammatory cascades, adipokines, mitochondrial dysfunction, and oxidative stress in hepatocytes as the second hit [12]. Subsequently, the existence of multiple risk factors was identified, including insulin resistance, lipotoxicity, inflammation, cytokine imbalances, lipid metabolism abnormalities, and genetic and epigenetic factors, leading to the proposal of a “multi-hit” hypothesis [15].

Besides the genetic polymorphism of PNPLA3 and TM6SF2, which leads to a reduction in VLDL secretion from hepatocytes and consequently to an increase in cellular TG concentration and lipid droplet content, in some cases, impairment of carbohydrate homeostasis related to incretins or alteration of pancreatic β-cell function is recognized as the primary cause for the development of MASLD, while in some individuals adipose tissue dysfunction, particularly visceral adipose tissue (VAT), has been identified [12].

Indeed, fat distribution is considered more important than excessive change in BMI [16], with VAT reflecting central obesity, which is characterized by ectopic fat accumulation in organs and is recognized as a metabolically active and inflammatory tissue that plays a critical role in the development of steatotic liver disease and MASLD. This may explain why, by impairing glucose and lipid metabolism and the function of several organs, including skeletal muscle and liver [17], higher levels of VAT are associated with greater metabolic risk, as well as hepatic inflammation and fibrosis, compared with peripheral and subcutaneous fat [16].

In lean, healthy individuals, VAT contributes 5–10% of portal blood FFAs, but this can reach 50% in individuals with elevated VAT. This may explain why higher levels of VAT are found in lean or NAFLD/MASLD patients without obesity compared to healthy controls [16]. In fact, lipid accumulation in the liver caused by de novo lipogenesis (DNL), free fatty acids released from adipose tissue and dietary fat, has been confirmed as the initial critical event of this condition [12]. DNL is a process that converts non-fatty acid substrates (glucose, lactate, and amino acids) to fatty acids and involves several enzymes, including acetyl-CoA carboxylase (ACC), fatty acid synthase (FAS), and stearoyl-CoA desaturase 1 (SCD1) [18].

Given the role of glucose and insulin in promoting DNL through activation of carbohydrate response element-binding protein (ChREBP) and sterol regulatory element-binding protein-1 (SREBP-1), this process is enhanced in the context of impaired glucose homeostasis and insulin resistance, which are sometimes associated with dietary patterns high in sugar and saturated fat, excess caloric intake, and reduced physical activity [19].

In fact, the imbalance between energy intake and insulin resistance in muscle, adipose tissue, and the liver, associated with a sedentary lifestyle and fat accumulation, is recognized as a critical factor for ectopic fat accumulation in individuals with MASLD. In this context, increased lipolysis, together with upregulated DNL in hepatocytes, leads to an excess of free fatty acids and glucose in the circulation, perpetuating insulin resistance and lipotoxicity and impairing beta-cell function. As a protective mechanism, adipose tissue attempts to store the excess fat, which increases with both hypertrophy and hyperplasia. This adaptation leads to dysfunction of adipose tissue with activation of macrophages and inflammatory pathways [20].

The inability of adipose tissue to store fat leads to the release of fat into the circulation via VLDL, which increases cardiovascular (CVD) risk but also alters adipokine release, such as adiponectin and leptin, which increase insulin resistance and can affect lipid homeostasis [9]. The metabolic stress associated with metabolic overload also depends on the individual’s ability to expand adipose tissue, which is influenced by many factors, including genetic, epigenetic, environmental, age, and gender [21].

The increase in free fatty acids, both from upregulated lipolysis, leads to their “unhealthy” storage in other tissues, such as liver, pancreas, or muscle, and their impaired oxidation. As a result, lipotoxic lipids are produced, leading to cellular stress, inflammation, tissue regeneration, and fibrogenesis, promoting a fibrogenic response in hepatic stellate cells that is responsible for the progression to MASH and ultimately cirrhosis [20].

Thus, in addition to insulin resistance, adipose tissue dysfunction plays a central role in the pathophysiology of fatty liver [1].

## 4. MASLD: From Adiposopathy to a Liver-Centered Approach

One of the most widely accepted theories explaining the progression of MASLD involves insulin resistance as the primary pathogenetic mechanism, which is closely linked to the presence of obesity. Insulin, a hormone secreted by pancreatic β-cells, plays a critical role in maintaining glucose homeostasis by promoting glucose uptake, glycogen synthesis, lipogenesis, and protein synthesis while inhibiting gluconeogenesis, lipolysis, apoptosis, and autophagy. In the presence of insulin resistance, hyperinsulinemia occurs, stimulating triglyceride synthesis and accumulation in the liver, a central organ involved in metabolic and homeostatic regulation.

It has been observed that approximately 30–90% of people with obesity, a condition strongly associated with insulin resistance, develop hepatic steatosis. Specifically, the hypertrophic expansion of VAT leads to a condition known as “sick fat” or “adiposopathy”, characterized by adipocyte hypoxia and an imbalance in the release of adipokines and cytokines that promote insulin resistance [16]. This process begins in the adipose tissue and subsequently affects the liver. In addition, the abnormal secretion of cytokines and adipokines increases lipolysis in adipose tissue through multiple signaling pathways, promoting the deposition of free fatty acids (FFAs) in muscle and liver, thereby exacerbating insulin resistance [17].

The ectopic deposition of FFA in the liver, characteristic of central obesity, induces a state of lipotoxicity that disrupts insulin signaling pathways and impairs hepatic metabolism. This condition results in decreased hepatic insulin secretion, increased glucose production, and increased production of triglyceride (TG)-rich lipoproteins such as VLDL (very low-density lipoprotein) [16], as well as downregulation of GLUT-4 expression. Dysfunctional adipose tissue also contributes to a state of low-grade inflammation characterized by upregulation of pro-inflammatory cytokines such as IL-6 and TNF-α and a decrease in the anti-inflammatory cytokine IL-10. These inflammatory changes inhibit insulin signaling, activate gluconeogenic enzymes, and exacerbate hepatic insulin resistance. In addition, this inflammatory state promotes oxidative stress and overproduction of reactive oxygen species (ROS). Oxidative stress causes direct damage to hepatocytes and promotes the progression of liver disease, a process supported by the increased lipid peroxidation observed in various models of MASLD. Thus, hepatic triglyceride synthesis in the setting of insulin resistance is recognized as a key mechanism in the pathogenesis of NAFLD/MASLD [17].

Recently, however, the existence of a bidirectional relationship between MASLD and insulin resistance has been proposed, with MASLD itself exacerbating insulin resistance. The basis of this liver-centric perspective is the ectopic accumulation of free fatty acids (FFAs) and triglycerides in hepatocytes. This accumulation occurs when the capacity of the liver to handle this “metabolic overload” is exceeded, resulting in metabolic–inflammatory stress that affects various end organs, including the liver itself [17,21]. This stress promotes insulin resistance through mechanisms such as oxidative stress, endoplasmic reticulum (ER) stress, lipotoxicity, and, in particular, dysregulated secretion of several hepatokines.

The altered secretion of hepatokines by hepatocytes in the development of MASLD has recently received increased attention. Because of their critical role in communication between the liver and target organs such as adipose tissue and muscle, impaired hepatokine signaling is thought to contribute significantly to metabolic dysregulation (Figure 2).

These findings support a shift from the two-hit theory to the multiple-hit theory of MASLD progression [17]. This updated theory suggests that the primary pathogenic mechanism originates in the liver and subsequently affects multiple extrahepatic organs, resulting in the metabolic abnormalities characteristic of MASLD.

## 5. The Analysis of Metabolic Organ-Secreted Factors in MASLD

The liver is increasingly recognized as a vital endocrine organ with a central role in the regulation of metabolism. It helps maintain energy homeostasis by managing energy storage and utilization under various metabolic conditions, such as exercise, fasting, diet, obesity, diabetes, metabolic syndrome, and liver dysfunction [22]. Because of its secretory capacity, the liver plays an essential role in crosstalk with other tissues, including the central nervous system, adipose tissue, and muscle [17]. Similar to the imbalanced secretion of adipokines from dysfunctional adipose tissue, with high leptin and lower adiponectin levels, a liver-centered perspective suggests that the metabolic dysregulation observed in MASLD is largely due to altered endocrine function of the liver, specifically an abnormal hepatokine secretion profile from hepatocytes [6,17]. Indeed, analogous to the molecular remodeling that occurs in adipose and muscle tissue due to ectopic lipid overload, researchers hypothesize that fatty liver disease alters the endocrine function of the liver compared to a healthy liver, with altered gene and protein expression that negatively affects cardiometabolic health [6,23]. Bioinformatic analyses of changes in protein secretion in steatosis predicted alterations in inflammation and metabolism, which were confirmed by cellular studies showing that the altered secretome profile of steatotic hepatocytes induces pro-inflammatory pathways and insulin resistance [23]. In addition, genomics of steatotic hepatocytes has suggested lipid droplet-associated gene regulation and perturbed metabolic pathways in the lipid load of human primary hepatocytes and human hepatoma cell culture, including in particular the PPAR family and FGF21. For example, a genome-wide meta-analysis found that the rs838133 variant in the FGF21 gene was associated with decreased protein and increased carbohydrate intake, which may contribute to the development of MASLD [24].

The hepatic proteome has been studied both in humans with NAFLD/MASLD and in animal models to better understand the pathophysiological mechanisms of liver steatosis [25]. The hepatic transcriptome and proteome in high-fat diet (HFD)-induced NASH/MASH mice have shown a reprogramming of the liver transcriptome and proteome [26]. For example, the expression of fibroblast growth factor (FGF21), which is also secreted by the liver, has been shown to be responsive to various signals, including starvation, certain nutrients, and various physiological and environmental stresses, and is increased in fatty liver [27]. Similarly, hepatic transcription of angiopoietin-like proteins 4 (ANGPTL4) is increased by fasting via PPARα signaling and is suppressed after refeeding [28].

A proteomic analysis of liver biopsies identified nearly 220 proteins with different levels in patients with NAFLD compared to obese metabolically healthy individuals. Proteins upregulated in NAFLD/MASLD were involved in PPAR signaling and extracellular matrix-receptor interactions, whereas proteins downregulated were mainly located in mitochondria and involved in oxidative phosphorylation [29].

The hepatic transcriptomes of MASLD also appear to differ from those of healthy subjects, and genes involved in immune regulation and extracellular matrix remodeling appear to be more characteristic of individuals with MASH than those with simple steatosis. Indeed, transcriptomic analyses have shown that MASLD, and especially MASH conditions, are characterized by upregulation of genes involved in tissue repair, cell adhesion and migration, extracellular matrix organization, immune function, and cancer progression, whereas genes involved in glucose metabolism, lipid metabolism, protein metabolism, oxidative stress, insulin signaling, mitochondrial function, and inflammation appear to be downregulated [25]. For example, an upregulation of inflammatory and lipid metabolic pathways has been observed in patients with severe NAFLD/MASLD, with overexpression of interleukin-32 and a positive correlation between this cytokine and the severity of steatosis [30]. In a validated mouse model of progressive NAFLD/MASLD, specific genes were shown to be differentially regulated during the development and progression of liver disease. In particular, an early increase in sterol regulator element binding protein-1 (SREBP-1), which is involved in de novo lipogenesis, was observed during disease development and its decrease during disease progression, despite the maintenance of an obesogenic diet and insulin resistance.

The progression of MASLD has been shown to be characterized by a concomitant increase in genes related to cytokines/chemokines, toll-like receptor and inflammasome, and genes involved in free fatty acid oxidation. This may represent a feedback mechanism leading to an increase in reactive oxygen species (ROS) production that remains elevated as the disease progresses [31]. This may explain why altered mitochondrial function, oxidative stress, ER stress, and autophagy have been implicated as drivers of hepatocyte apoptosis in NASH/MASH, which is recognized as a critical aspect of progressive NAFLD/MASLD [32,33].

Taken together, these findings demonstrate that there are numerous differences in the activation of inflammatory pathways and cellular stress in the early phase of the disease compared to the advanced phase and that hepatokines are likely involved not only in metabolic regulation but also in the development and progression of metabolic diseases such as obesity, T2D, MASLD, and MASH.

## 6. The Role of Hepatokines in the MASLD Metabolic Dysfunction

Several key hepatokines have been identified, including fetuin-A, angiopoietin-like protein (ANGPTL), fibroblast growth factor 21 (FGF21), insulin-like growth factors (IGF), selenoprotein P (SeP), leukocyte-derived chemotaxin 2 (LECT2), and sex hormone-binding globulin (SHBG). Like other organokines such as adipokines and muscle-derived myokines, hepatokines are hormone-like proteins secreted primarily or exclusively by the liver into the circulation and act in an endocrine or paracrine manner [34]. These molecules serve as pleiotropic signaling agents capable of providing information about the metabolic status of the liver [35]. They have a positive or negative regulatory effect on the pathogenesis of MASLD by influencing glucose and lipid metabolism, oxidative stress, and systemic inflammation [36]. Therefore, they have been implicated in the development of obesity, insulin resistance, and hepatic steatosis [37], and their upregulation in patients with T2D has been shown to contribute to the development of insulin resistance, which may explain why MASLD often precedes dysfunction in other organs, leading to systemic metabolic disease [28].

In support of this theory, studies using hepatocytes isolated from both healthy and steatotic mice have shown that liver steatosis impairs hepatokine secretion, which can subsequently disrupt lipid metabolism, induce inflammation, and lead to insulin resistance in other tissues [35]. Xiong et al. demonstrated that the transcriptome and secretome of high-fat diet-induced (HFD) NASH mice were completely reprogrammed, resulting in the secretion of hepatokines. These hepatokines have the potential to modulate energy metabolism and promote the progression of metabolic diseases such as obesity, insulin resistance, T2D, NAFLD/MASLD, and NASH/MASH [22,38].

The changes in hepatokine secretion associated with the onset of NAFLD/MASLD may affect inter-organ communication [28], thereby impairing nutrient metabolism in the liver and other peripheral tissues [9]. This dysregulated signaling may contribute to the progression of various metabolic disorders by promoting insulin resistance in the liver, muscle, adipose tissue, and pancreas [17]. In addition, MASH may be considered an endocrinopathy mediated by liver-produced hepatokines that affect metabolic regulation beyond the liver [39].

Although only a few hepatokines have been identified as contributors to the progression of liver steatosis, understanding how hepatokines drive inter-organ communication is critical to understanding the complex metabolic networks between tissues. This knowledge may also help to identify new diagnostic and therapeutic targets for metabolic diseases [28]. This review will analyze the role of several hepatokines in influencing the development and progression of MASLD (Table 1).

For example, ANGPTL appears to promote MASLD by affecting lipid metabolism and inflammatory response, LECT2 may impair insulin resistance, inflammation, and lipid metabolism, and SeP may induce lipid dysregulation. Special attention will be paid to the two hepatokines fetuin-A and FGF21, which are the most likely to be the most studied hepatokines in humans and animals [6]. They appear to play opposing roles in the development of MASLD, with fetuin-A promoting insulin resistance and inflammation and consequently the development and progression of MASLD, whereas FGF21 appears to have protective effects by regulating lipid and glucose homeostasis and fatty acid oxidation [39]. Therefore, their modulation may be considered as a possible therapeutic target in MASLD (Figure 3).

## 7. Angiopoietin-like Protein (ANGPTL)

Angiopoietin-like proteins (ANGPTL 1–8) are glycoproteins involved in the regulation of lipid metabolism. Among them, ANGPTL3 is exclusively released by hepatocytes with positive influence on expression by liver X receptor (LXR) and negative regulation by insulin, leptin, peroxisome proliferator-activated receptor-β, statins, and thyroid hormones [6]. This protein plays an important role in several physiopathological conditions, including energy homeostasis, tumorigenesis, angiogenesis, and redox regulation [28]. Indeed, ANGPTL3 is able to inhibit the activity of lipoprotein lipase (LPL), which improves lipid metabolism and insulin resistance, promoting the storage of triglycerides in the gluteo-femoral compartment, with a protective effect on fat accumulation in visceral fat [6]. Therefore, this inhibitory activity results in upregulation of plasma triglycerides, increased storage of fatty acids in white adipose tissue, and increased insulin resistance, contributing to the cardiovascular risk factors associated with LPL inhibition [28,37].

A positive correlation was found between ANGPTL3 and plasma glucose, insulin, and HOMA-IR levels in patients with insulin resistance [37], and its levels were higher in patients with NASH/MASH than in patients with simple steatosis [40,41]. Therefore, inhibition of ANGPTL3 has a protective effect on insulin resistance, lipid metabolism, and hepatic steatosis. Selective pharmacotherapeutic inhibition of ANGPTL3 synthesis in patients with T2D, hypertriglyceridemia, and hepatic steatosis significantly lowered triglyceride levels compared to placebo without improving glycemic parameters or hepatic fat content [59]. Since studies have shown that obese mouse models treated with an ANGPTL3 antagonist peptide showed a marked reversal of diet-induced obesity and hepatic steatosis [28], the lack of improvement in glycemia and steatosis in the previous study has been attributed to underpowered effects due to its small sample size [6]. ANGPTL4 is predominantly expressed in adipose tissue and liver and regulates energy homeostasis and triglyceride metabolism. Although it appears to be increased in people with T2D, its serum concentrations in insulin resistance are still controversial [37]. It enhances lipolysis in adipose tissue and inhibits LPL activity, thereby downregulating the clearance of triglycerides in the blood. Overexpression of ANGPTL4 has been associated with a reduction in adipose tissue, increased blood levels of triglycerides, FFA, and cholesterol, and consequently the risk of hepatic steatosis, while its role in the regulation of glucose metabolism remains unresolved and is considered an important area for future research [23]. Recent studies have reported that lack of ANGPTL4 expression in a mouse model decreased triglyceride and cholesterol levels, improved glucose tolerance, and reduced atherogenesis [42,43]. ANGPTL6 is also mainly secreted by the liver, and a higher incidence of obesity, diabetes, fat accumulation in liver and muscle, and insulin resistance have been observed in mice lacking this protein. In contrast, overexpression of ANGPTL6 increases insulin sensitivity and energy expenditure and protects against the development of hepatic steatosis. Its ability to increase adenosine monophosphate-activated protein kinase (AMPK) expression in muscle and decrease gluconeogenesis in the liver has been observed, suggesting a protective role in energy and glucose homeostasis [37]. As studies have shown higher levels of this protein in patients with obesity and T2D, the existence of ANGPTL6 resistance in these conditions has been suggested [34,37,60].

## 8. Insulin-like Growth Factors (IGFs)

Insulin-like growth factors (IGFs) are expressed in several tissues, but most of the circulating IGF-I comes from the liver, so it can be considered a hepatokine [60]. They are related to insulin in both structure and function and play an important role in improving insulin sensitivity.

In fact, IGF1 is released by the liver and acts primarily on muscle, where it improves insulin sensitivity [37]. It has been hypothesized that lower levels of IGF1 may predict the development of diabetes, insulin resistance, metabolic syndrome, and cardiovascular disease. Its levels have been found to be reduced in patients with NAFLD/MASLD and obesity, with data suggesting that hepatic insulin resistance may influence IGF-I levels through modulation of growth hormone-stimulated hepatic synthesis of IGF-I [37,60]. Although it is clear that this factor contributes to the maintenance of insulin activity, the overall mechanism remains to be elucidated [37].

## 9. Selenoprotein P (SeP)

Selenoprotein (SeP) is a glycohepatokine that is abnormally regulated in hepatic steatosis. SeP interferes with the insulin pathway in liver and muscle and inhibits the activation of AMPK. Its expression is enhanced by higher palmitate and glucose levels, increasing insulin resistance and impairing β-cell function, whereas insulin downregulates its expression [60]. In fact, treatment with high SeP has been shown to decrease the volume of α- and β-cells [44]. SeP expression has been found to be upregulated in patients with NAFLD/MASLD and/or T2D, suggesting its correlation with insulin resistance, hyperglycemia, and inflammation [6,23]. These findings are consistent with studies in mice showing increased synthesis of SeP in response to high-fat diets, NAFLD/MASLD, and T2 [37,45]. In addition, SeP has been shown to correlate with cardiovascular risk factors in patients with cardiovascular disease, including waist circumference, triglycerides, and carotid intima-media thickness [61]. Furthermore, SeP levels are negatively related to adiponectin levels in patients with T2D [60], suggesting its possible role in crosstalk with the other organokines.

The observation of improvement of insulin resistance and protection of islets and insulin secretion after treatment with antibodies against SeP suggests that targeting this protein may be a new approach to treat MASLD and various metabolic disorders [44]. Indeed, the alteration of SeP in several components of metabolic disorders makes it a potential biomarker in the prediction of MASLD [61].

## 10. Leukocyte Cell-Derived Chemotaxin 2 (LECT2)

Leukocyte cell-derived chemotaxin 2 (LECT2) is mainly synthesized by the liver but also by adipose tissue, neurons, and white blood cells. Its synthesis is highly sensitive to changes in dietary fat content as well as to the severity of hepatic steatosis [28]; indeed, LECT2 has been shown to be dysregulated in the presence of hepatic steatosis, and its upregulation is associated with advanced stages of human liver fibrosis [46]. Consistent with this, findings from animal studies also suggest that LECT2 exacerbates fibrosis and impairs insulin signaling in muscle and adipose tissue [6,47,62]. In humans, visceral fat area has been identified as the strongest predictor of plasma LECT2, which is a potential biomarker linking visceral obesity to dyslipidemia [6,63]. In addition, it has been suggested that the association of LECT2 with abdominal obesity and lipid metabolism may influence its relationship observed with both MASLD and metabolic syndrome [64]. Overall, further studies are needed to investigate whether LECT2 can be used as a potential target for the treatment of MASLD-related insulin resistance [6].

## 11. Sex Hormone-Binding Globulin (SHBG)

Because sex hormone-binding globulin (SHBG) is predominantly secreted by hepatocytes, its role in the regulation of human metabolism has been recognized [60]. The level of circulating SHBG has been associated with the ability to influence glucose metabolism, amount of adipose tissue, metabolic disorders, and lipid content in the liver [34]. This may explain why its levels have been negatively associated with obesity, T2DM, metabolic syndrome, and hepatic steatosis [44]. Tumor necrosis factor alpha (TNFα) and the interleukin IL1β downregulate SHBG expression, whereas adiponectin upregulates its expression [6]. SHBG expression has been linked to PI3K/AKT pathway function in a human cellular model of insulin resistance [65] and to inhibition of hepatic lipogenesis in mouse models by reducing PPARγ activation [48]. Furthermore, this globulin has been shown to protect against ER stress and its progression [6].

For example, it has been observed that adolescents with obesity and lower SHBG were more likely to develop NAFLD/MASLD, with a negative correlation between SHBG and several parameters, including blood pressure, BMI, waist circumference, insulin, and Homeostatic Model Assessment of Insulin Resistance (HOMA-IR) [66].

According to recent studies, overexpression of SHBG downregulated lipogenesis and reduced hepatic steatosis, suggesting its protective role in NAFLD/MASLD [48,67]. Therefore, further studies are needed to elucidate the role of SHBG in MASLD.

## 12. Fibroblast Growth Factor 21: Structure and Function

Fibroblast growth factors (FGFs) are a family of peptide factors involved in tissue repair and regeneration and in the regulation of metabolic homeostasis through their interaction with FGF receptors. Fibroblast growth factor 21 (FGF21) is a 209 amino acid protein mainly secreted by the liver [28,34], which can also be expressed in various organs, including muscle, white and brown adipose tissue, intestine, heart, kidney, and pancreas [49]. As a peptide hormone, FGF21 activates cell signaling in target tissues by binding to the heteromeric complex between fibroblast growth tyrosine kinase receptors (FGFRs), particularly FGFR1c, and the co-receptor β-klotho (KLB) [68]. Co-expression of these receptors determines the sensitivity of tissues or organs to FGF21 [49] and allows FGF21 to target the liver itself, as well as the pancreas and adipose tissue [69]. In fact, as an endocrine factor, this epatokine acts on multiple tissues beyond the liver, including adipose tissue, muscle, pancreas, and other organs to regulate metabolic functions [70].

When FGF21 binds to the FGFR-KLB dimer, it stimulates phosphorylation of FGFR substrate 2α (FRS2α) and activates extracellular signal-regulated kinase 1/2 (ERK1/2) and Akt [71]. Accumulating evidence suggests that it plays an important protective role in lipid and carbohydrate metabolism in an insulin-independent manner and in maintaining energy homeostasis [34,50], with anti-obesity, anti-diabetic, and anti-hyperlipidemic effects [70].

Regarding its secretion, hepatic FGF21 expression is tightly regulated during starvation by peroxisome proliferator-activated receptor α (PPARα), a transcription factor activated by non-esterified fatty acids released from adipocytes. PPARα can induce FGF21 expression in the liver, leading to decreased lipogenesis and increased fatty acid β-oxidation [50]. Similarly, in adipose tissue, FGF21 expression and function are regulated by peroxisome proliferator-activated receptor-γ (PPARγ) after feeding. PPARγ is a transcription factor that exerts systemic insulin-sensitizing effects by inducing the expression of genes involved in the insulin signaling cascade and the synthesis of adipokines such as adiponectin. On the other hand, FGF21 can also increase the expression and activity of PPARα and PPARγ, thereby controlling energy metabolism and oxidative stress in various tissues.

Among its beneficial functions in human metabolism, FGF21 improves insulin sensitivity and glucose absorption by activating the phosphatidylinositol 3-kinase (PI3K)/Akt signaling pathway and promoting the expression of GLUT-4 [51]. In white adipose tissue, this epatokine activates extracellular signal-regulated kinases (ERK1/2) and stimulates the expression of insulin-independent glucose transporter 1 (GLUT-1), which increases glucose uptake and improves insulin sensitivity [50]. In addition, there appears to be a synergistic effect between insulin and FGF21, with insulin enhancing the regulatory effect of FGF21 on GLUT-1 expression and FGF21 enhancing the hypoglycemic effect of insulin. FGF21 also improves insulin resistance by acting directly on islet β-cells by increasing the number of insulin receptors, inhibiting glucolipotoxicity-induced apoptosis, increasing insulin secretion, and reducing α-cell glucagon secretion [51]. However, these mechanisms do not fully explain the hypoglycemic effects of FGF21. A significant portion of its hypoglycemic activity is mediated by its ability to induce adiponectin expression through activation of the PPARγ pathway in white adipose tissue. Indeed, increased plasma adiponectin levels have been observed following administration of FGF21 or its analogs. This adipokine, known for its insulin-sensitizing, anti-inflammatory, and anti-sclerotic effects, plays a key role in mediating the effects of FGF21 on energy metabolism and insulin sensitivity, increasing glucose transport, and lowering blood glucose [50].

FGF21 also plays an important role in lipid metabolism. It helps reduce blood levels of total cholesterol, low-density lipoprotein (LDL), and triglycerides (TG), while increasing plasma levels of high-density lipoprotein (HDL) [51]. FGF21 is also essential for hepatic lipid metabolism and prevents the development of steatohepatitis by increasing fatty acid oxidation and turnover in the liver [72,73]. For example, upregulation of FGF21 has been observed as a compensatory mechanism in hepatic steatosis associated with long-term alcohol exposure. In addition, inhibition of hepatic fatty acid β-oxidation occurs in the absence of the FGF21 gene, as demonstrated in knockout models [51]. FGF21 is also involved in maintaining energy balance by inducing the expression of uncoupling protein 1 (UCP1) and other thermogenesis genes, thereby promoting browning of white adipose tissue (WAT) [52].

Finally, although it is not known whether the brain can release FGF21, it has been suggested that this hormone may play a role in regulating metabolism by crossing the blood–brain barrier and binding to FGFRs and β-klotho expressed in the hypothalamus [51]. It has been shown to regulate simple sugar intake and sweet taste preference in response to activation of carbohydrate responsive element-binding protein (ChREBP) acting on its receptors in the paraventricular nucleus of the hypothalamus. It then reaches its maximum levels in response to a diet low in protein and high in carbohydrates, allowing it to exert its protective effects [37].

The complex of these metabolic activities confirms the pleiotropic activity of FGF21, improving insulin sensitivity, hepatic steatosis, and body weight and leading to its recognition as a potential therapeutic target for obesity-related metabolic disorders, including NAFLD/MASLD [53]. This may explain why the analogs or activators of this hepatokine are classified as anti-obesity and anti-diabetic molecules.

## 13. MASLD: An Alteration of FGF21 Expression

Clinically, elevated circulating levels of FGF21 have been observed in several dysmetabolic conditions, including obesity, insulin resistance, and T2D [37]. Furthermore, since FGF21 is produced by hepatocytes, it is reasonable to assume that liver disease could affect its expression, with elevated FGF21 levels observed in MASH patients with obesity [24]. In particular, plasma levels of FGF21 have been shown to increase in parallel with the severity of hepatic steatosis in humans [74,75,76], and a positive correlation has been observed with the severity and progression of MASLD, as well as with obesity, body mass index (BMI), triglycerides, and insulin resistance [53]. In fact, it appears that chronic trygliceride accumulation in the liver creates a stress condition that damages hepatocytes, leading to increased transcription and secretion of FGF21 [70], suggesting it as a potential biomarker for metabolic disorders [17]. Since this hormone plays several metabolic activities such as decreasing lipolysis, increasing fatty acid oxidation in adipose tissue, decreasing hepatic lipid levels such as diacylglycerol, increasing insulin sensitivity, and improving glycemic control, the elevated circulating FGF21 levels observed in people with NAFLD have been suggested by several studies to be an adaptive mechanism to maintain energy homeostasis under metabolic stress [35]. Indeed, increased secretion of FGF21 has been hypothesized to occur in the context of increased fatty liver and carbohydrate signaling [6]. This is consistent with the observation that the effects of FGF21 are significantly enhanced in response to metabolic, oxidative, nutritional, hormonal, or environmental stress, underscoring its critical role in restoring metabolic homeostasis [49]. However, despite the beneficial effects of FGF21, its elevated levels in human chronic metabolic diseases appear to be insufficient to improve these conditions [70].

It has been hypothesized that obesity and insulin resistance may lead to a phenomenon similar to insulin or leptin resistance, known as FGF21 resistance, which is considered a self-protective mechanism of the body. This condition could explain why mice with severe hepatic insulin resistance, due to the absence of hepatic insulin receptor substrates (IRSs) IRS1 and IRS2, exhibit reduced obesity and lower circulating levels of FGF21 [6].

In addition to this impaired endocrine signaling of FGF21, which may be due to a decrease in FGF21 receptor expression or function, there is also evidence of significant upregulation of fibroblast activation protein (FAP) in the liver and serum of patients with metabolic liver disease. FAP is an endopeptidase that inactivates endogenous FGF21, further contributing to FGF21 dysfunction [70].

Overall, these findings have guided the design and development of several FGF21 analogs and mimetics that have shown promising therapeutic efficacy against obesity and T2D, demonstrating anti-inflammatory, antidiabetic, and hypolipidemic properties [28,49]. This could explain why studies have found that pharmacological interventions based on FGF21 analogs have reduced hepatic steatosis and improved several biomarkers of liver fibrosis in patients with NASH [54,77]. Consequently, these compounds have been classified not only as anti-obesity and anti-diabetic agents but also as potential treatments for steatotic liver disease [24].

## 14. MASLD: Potential Therapeutic Action of FGF21

There is a clear need for effective treatments for MASLD, underscoring the importance of developing platforms to test novel compounds [24]. While incretin-based therapies have demonstrated significant effects in the treatment of obesity and its complications, other therapeutic targets are being explored. In particular, despite the paradoxically elevated expression of FGF21 in people with NAFLD/MASLD, which correlates with liver histopathology [78], its beneficial effects on energy homeostasis through hypothalamic actions, as well as direct benefits on adipose tissue (e.g., improvement of insulin sensitivity) and the liver (reduction in lipid overload), have led to considerable efforts in the last decade to develop FGF21 derivatives or specific FGF21 agonists for the treatment of various metabolic disorders, including T2D, obesity, and NAFLD [79]. For example, genomic studies of fat-loaded hepatocytes have revealed alterations in lipid droplet-related gene regulation and disrupted metabolic pathways, including those involving the PPAR family and FGF21 [24]. The promising metabolic benefits of FGF21 that contribute to the resolution of hepatic steatosis are supported by evidence from rodent models of NAFLD/MASLD and NASH/MASH, where FGF21 analogs have reduced hepatic steatosis. These benefits are also attributed to the extrahepatic actions of FGF21.

Steatosis in hepatocytes is characterized by the microscopic accumulation of triglycerides within cytosolic lipid droplets (LDs). The lipotoxic effects are not directly caused by the accumulation of esterified triglycerides in these LDs, but rather by the release of lipotoxic species, such as free fatty acids, during LD turnover. Saturated fatty acids released from LDs or taken up from the circulation can directly cause tissue injury, potentially leading to further recruitment and activation of macrophages, which exacerbate liver damage. FGF21, by interacting with its primary receptor FGFR1c expressed in adipose tissue, appears to suppress the flux of FFAs from adipose tissue to the liver. This mechanism, together with increased fat oxidation, may contribute to the reduction in hepatic steatosis, including that associated with NASH. Additional mechanisms that contribute to the reduction in hepatic steatosis include suppression of caloric intake, inhibition of de novo lipogenesis in the liver, and increased insulin sensitivity in adipose and muscle tissue, which promotes glucose uptake in peripheral organs [70]. Indeed, FGF21 reduced hepatic levels of the transcription factor SREBP-1, decreased hepatic lipogenesis, increased hepatic and adipose tissue expression of the metabolic coactivator PGC-1α, and appeared to protect β-cell function in diabetic mouse models [60]. Given these effects, other MASLD therapeutic targets, such as FGF21 analogs, are also being investigated in ongoing research [80].

Several of these analogs have demonstrated in both preclinical and human studies the ability to not only act directly on the liver but also to exert pleiotropic effects that improve liver health and restore whole-body metabolism [70].

In diet-induced obese mice, FGF21 analogs have been reported to decrease the expression of lipogenic genes typically upregulated by a high-fat diet and to increase the expression of genes involved in mitochondrial β-oxidation of fatty acids [17]. In addition, these analogs have been shown to reduce body weight, liver and circulating triglycerides, fasting plasma insulin, and glucose levels while increasing energy expenditure in obese or metabolically induced animals [70].

Clinical trials have evaluated these compounds as potential pharmacological interventions for patients with obesity, T2D, and NASH. In patients with obesity and T2D, treatment with long-acting FGF21 analogs resulted in significant improvements in body weight and circulating lipoprotein levels. However, the beneficial effects on glucose and insulin levels did not reach statistical significance, leaving the precise role of FGF21 in human glycemic control unclear [81,82]. Regarding lipid metabolism, FGF21 analogs have significantly reduced plasma triglyceride concentrations and increased HDL cholesterol levels by reducing non-esterified fatty acids and increasing lipoprotein catabolism in both white and brown adipose tissue [54,81,82,83]. These findings suggest that FGF21-based therapies may be effective in the treatment of metabolic disorders, including NASH. Indeed, FGF21 analogs have demonstrated a favorable pharmacological profile in reversing non-alcoholic fatty liver disease and its more severe stages, such as steatohepatitis. A phase 2 study showed significant improvements in liver fat and blood biomarkers of fibrosis in patients with NASH treated with PEGylated FGF21 [6,77]. Given that FGF21 is recognized as a promising biomarker for the diagnosis and staging of NAFLD and that its elevated levels in patients with hepatic steatosis may indicate a protective response to lipotoxicity, research suggests that exogenous FGF21 could potentially slow the progression of these diseases. According to the “multiple strikes” theory of NAFLD/MASLD pathogenesis, FGF21 influences several other critical processes, including oxidative stress, endoplasmic reticulum stress, mitochondrial dysfunction, and low-grade inflammation, which could help attenuate the progression of liver steatosis [34].

In addition to its multiple actions that collectively reduce intrahepatic lipid accumulation, such as increasing hepatic insulin sensitivity, reducing hepatic de novo lipogenesis, and increasing mitochondrial fatty acid β-oxidation, FGF21 also activates other mechanisms that contribute to the resolution of hepatic steatosis, inflammation, and possibly fibrosis. FGF21 attenuates hepatic inflammation by inhibiting the NF-κB pathway and enhancing Nrf2-mediated antioxidant capacity. These effects result in decreased levels of pro-inflammatory cytokines such as IL-1β, increased levels of anti-inflammatory cytokines such as IL-10, and reduced intrahepatic endoplasmic reticulum (ER) stress [55]. In addition, FGF21 downregulates VLDL receptor expression in hepatocytes, thereby reducing the delivery of very low-density lipoprotein (VLDL) to the liver. It also lowers postprandial triglyceride levels and promotes fatty acid storage in adipose tissue [50,53].

Preclinical studies with FGF21 analogs have shown promising effects, including inhibition of inflammation and immune cell infiltration in the liver, reduction in liver injury and hepatocyte death, significant improvement in liver fibrosis, and prevention of NASH/MASH development [51]. For example, three weeks of treatment with FGF21 and the FGF21 analog LY2405319 reduced oxidative stress and liver weight in a NASH mouse model [22].

Although the evidence is limited and more research is needed, there is also evidence that FGF21 analogs may potentially reduce liver fibrosis, and others are being evaluated to assess their efficacy, safety, and tolerability in patients with cirrhosis [84]. In fact, studies in patients with MASH and fibrosis have shown the efficacy of some FGF21 analogs in improving fibrosis as well as resolving MASH [85,86], and other studies are ongoing in patients with cirrhosis [84]. Importantly, the beneficial effects of FGF21 in improving NASH/MASH and other comorbidities are attributed not only to its direct effects on hepatocytes but also to its ability to induce adiponectin production in adipose tissue or through a brain–liver axis [6]. Adiponectin, known for its anti-diabetic and insulin-sensitizing properties, directly reduces hepatic steatosis and inflammation and may also reduce hepatic stellate cell activation and migration. The beneficial activities of adiponectin are often reduced in patients with NASH/MASH, likely due to decreased expression of its receptors in the liver. FGF21 analogs have been shown to increase adiponectin levels in non-human primates and humans with metabolic diseases such as T2D, obesity, and NASH. This increase in adiponectin may mediate the indirect effects of FGF21 analogs by enhancing peripheral energy intake and suppressing hepatic steatosis and inflammation [70].

Overall, the efficacy of FGF21 in the treatment of metabolic disorders needs to be validated in large, multi-center trials in the future [34].

## 15. Fetuin-A: Structure and Function

Fetuin-A, also known as alpha-2-Heremans–Schmid glycoprotein (AHSG), is a heterodimeric globular protein composed of two subunits: a heavy A chain of approximately 282 amino acids and a light B chain of 12 amino acids, linked by half-cystine residues within their amino acid sequences [87]. This protein, which belongs to the cystatin superfamily of protease inhibitors, is encoded by the AHSG gene. Serum concentrations of fetuin-A are significantly higher in fetuses than in adults, with levels typically ranging from 300 to 1000 μg/mL in healthy adults of both sexes and being 5–50 times higher in fetal blood.

Primarily synthesized by hepatocytes (over 95%), fetuin-A is recognized as a pleiotropic glycoprotein involved in various physiological and pathological processes. Its ability to bind multiple receptors, including those for insulin, growth hormone, nerve growth factor (NGF), platelet-derived growth factor (PDGF), transforming growth factor (TGF)-II, TGF-β2, basic fibroblast growth factor (bFGF), and several toll-like receptors (TLRs), allows it to regulate bone remodeling, calcium metabolism, endocytosis, brain development, and the transport of metals and small molecules, including fatty acids, in the bloodstream. In addition, fetuin-A plays a critical role in the regulation of fatty acid and protein metabolism and in the modulation of both anti-inflammatory and pro-inflammatory processes [56].

The activity of fetuin-A provides important insights into the role of the liver in regulating glucose and lipid metabolism through the release of various hepatokines, thereby strengthening their association with obesity, T2D, and hepatic steatosis. However, the exact mechanisms underlying these associations remain unclear [56,69,87]. Unlike adipocytokines, which are produced by fat cells, fetuin-A is primarily synthesized in the liver. This may explain its elevated levels in people with obesity and insulin resistance, as well as in patients with chronic kidney disease (CKD) and end-stage renal disease (ESRD), where it is associated with an increased risk of developing diabetes [88].

In glucose regulation, fetuin-A was first identified as a physiological inhibitor of insulin receptor tyrosine kinase in liver and muscle [89]. In addition, phosphorylated fetuin-A was found to inhibit the glucose transporter type 4 (GLUT-4) translocation, thereby reducing glucose uptake and glycogen synthesis in skeletal muscle cells [57]. The role of fetuin-A in impairing insulin sensitivity is further supported by the fact that fetuin-A knockout (KO) mice are protected from weight gain and insulin resistance [58]. In addition, fetuin-A contributes to lipid-induced insulin resistance by promoting the accumulation of triacylglycerol in hepatocytes and increasing the expression of sterol regulatory element-binding protein 1c (SREBP-1c).

Fetuin-A also exacerbates adipose tissue inflammation and the development of insulin resistance by increasing free fatty acid binding to toll-like receptor 4 (TLR-4) [28], while decreasing adipose tissue PPARγ and adiponectin expression, further reducing its anti-inflammatory and insulin-sensitizing effects. Given that fetuin-A has been recognized as a direct contributor to the onset of insulin resistance in both humans and animals, researchers have extensively documented the concomitant increase in insulin resistance and obesity associated with higher levels of fetuin-A [87]. In addition, fetuin-A has been shown to promote cytokine expression and low-grade inflammation by activating several pathways potentially involved in the development of NAFLD/MASLD, insulin resistance, beta cell apoptosis, and T2D. Although suggested by in vitro apoptosis data, a complete characterization of islet inflammation and inflammatory cytokine levels is lacking [56,90]. It also affects energy homeostasis by interfering with the silent information regulator 1 (SIRT1) and AMP-activated protein kinase (AMPK) sensors [37,91].

Taken together, these diverse effects of fetuin-A on the development and progression of various pathologies suggest that this hepatokine could be considered both a biomarker and a target for the diagnosis and treatment of associated diseases [56].

It is important to note that some data from Mendelian randomization analyses confirm the causal relationship between fetuin-A and both cardiovascular and T2D risk, while others do not recognize this causal relationship. Therefore, these results must be interpreted with caution due to the heterogeneity of the populations included [6]. In fact, it seems that fetuin-A has both anti-inflammatory and pro-inflammatory effects depending on the stimulus of activation in different clinical conditions. For example, although it is not yet clear which effect is dominant, in contrast to liver-derived fetuin-A with pro-inflammatory effects, kidney-derived fetuin-A has been shown to play a local role in protecting renal integrity against hypoxia-induced chronic and progressive renal injury [56]. Furthermore, despite its negative effects on insulin sensitivity and lipid metabolism, fetuin-A is a known inhibitor of vascular calcification, which may suggest a protective effect counteracting its pro-inflammatory activity [92]. However, a correlation between serum fetuin-A levels and carotid atherosclerosis has been observed, suggesting its role as an atherogenic factor rather than an inhibitor of vascular calcification. Again, it seems to have a biphasic role in cardiovascular etiology, which may depend on the stage of atherosclerosis but also on the level of this hepatokine [56]. Taken together, this suggests that further studies are needed to fully understand all the multiple functions of fetuin-A in humans.

## 16. Fetuin-A: A Potential Biomarker and Pathogenetic Mechanism for Metabolically Associated Steatotic Liver Disease (MASLD)

Although the exact molecular mechanisms are not fully understood, there is strong evidence that fetuin-A plays a critical role in the pathophysiological pathways leading to various metabolic disorders, making it a valuable target for early diagnosis in clinical practice. Indeed, elevated fetuin-A levels have been positively correlated with markers of early atherosclerosis, including metabolic syndrome, obesity, insulin resistance, and low-grade inflammation, with a particularly strong association observed in hepatic steatosis. In addition, an inverse relationship between fetuin-A and adiponectin concentrations has been observed. Given that obesity and insulin resistance are major risk factors for T2D, this may explain why higher circulating fetuin-A levels have also emerged as a strong predictor of incident T2D [87].

The abnormal secretion of fetuin-A is thought to be triggered by lipid accumulation in adipose tissue combined with elevated blood glucose levels, possibly through activation of the extracellular signal-regulated kinase 1/2 (ERK1/2) and NF-κB pathways [56]. A study of individuals with both NAFLD/MASLD and T2D has shown a positive association between serum fetuin-A levels and body weight, BMI, and waist circumference [57,93], although the results must be interpreted with caution due to the small study group. This could explain the observed association between plasma fetuin-A and insulin resistance in observational studies [89,94,95]. However, some limitations must be considered, such as the potential for residual confounding, which is consistent with observational studies, and that it cannot be excluded that other factors or imprecision in the measurement of covariates may have influenced these observations [95]. This is further supported by evidence of increased fetuin-A expression in conditions such as fatty liver, obesity, and T2D, as well as in animals fed a high-fat diet [56,87]. Elevated levels of fetuin-A in adults have been independently associated with ectopic fat accumulation in the liver and morphologically confirmed NAFLD/MASLD conditions [93,96], while lower levels are observed in people with reduced liver fat [56,87]. A recent retrospective study also evaluated the levels of fetuin-A in a group of fifty patients diagnosed with MASLD compared to fifty healthy controls and found that fetuin-A was higher in the cases group, with a significant correlation with ultrasound grading (US), fibroscan with controlled attenuated parameter scan (CAP scan), and MASLD scores. In particular, considering the CAP score, fetuin-A levels were significantly higher in S2 and S3 cases than in S1, as well as in US grade 3 cases than in grades 1 and 2. These findings have led the authors to its recognition as a significant predictive factor of MASLD with high sensitivity, specificity, and accuracy, and its crucial role not only in the diagnosis but also in the assessment of the severity of MASLD. However, this study included a relatively small sample size, and the impact of serum fetuin-A on patient prognosis should have been evaluated [97].

Another study, behind observing a correlation with several metabolic parameters and with moderate-severe NAFLD/MASLD in young adults, independent of confounders, has also found that triglyceride levels were independent predictors of fetuin-A levels, confirming its role in directly affecting the metabolism of triglycerides and its accumulation. However, the small sample size in this study may have influenced the significant correlation of fetuin-A only with moderately severe NAFLD and has assessed the presence and severity of NAFLD/MASLD presence and severity of NAFLD using ultrasound without confirmation by the gold standard for its diagnosis, which is liver biopsy [98]. Indeed, liver expression of the fetuin-A gene has been found to correlate with the expression of key enzymes involved in glucose and lipid metabolism, such as carnitine palmitoyltransferase 1 (CPT-1), sterol regulatory element-binding protein 1c (SREBP-1c), fatty acid synthase (FAS), phosphoenolpyruvate carboxykinase 1 (PEPCK-1), and glucose-6-phosphatase (Glu-6-P) [87]. Fetuin-A overexpression promotes fat accumulation in adipocytes and in the liver by stimulating the mammalian target of rapamycin (mTOR), which in turn increases the expression of SREBP-1c and upregulates genes involved in de novo lipogenesis. This fat accumulation in NAFLD/MASLD further exacerbates the upregulation of fetuin-A levels [56].

Recent studies have highlighted the crucial interplay between the liver and adipocytes, mediated by the crosstalk between two serum proteins: fetuin-A and adiponectin [88]. Fetuin-A is thought to contribute to insulin resistance through multiple mechanisms, including impairment of insulin receptor signaling, inhibition of GLUT-4 translocation, and reduction in PPARγ and adiponectin expression, as shown in obese insulin-resistant mice [99]. Adiponectin is known for its insulin-sensitizing properties and its ability to mitigate the adverse effects of inflammatory mediators. Individuals with NAFLD/MASLD have been found to have lower serum adiponectin levels compared to healthy individuals, with these lower levels correlating with the severity of fibrosis, steatohepatitis, and liver inflammation. Fetuin-A, secreted by the liver, has been shown to inhibit adiponectin production in adipose tissue by suppressing its mRNA expression in cultured human adipocytes [88]. In addition, fetuin-A mRNA expression is elevated in individuals with NASH/MASH compared to those with simple fatty liver [87], where it negatively regulates adiponectin, thereby impairing its insulin-sensitizing and anti-inflammatory effects. The decrease in adiponectin levels leads to proliferation of hepatic stellate cells and generation of reactive oxygen species in the liver, driving the progression from hepatic steatosis to steatohepatitis and eventually to cirrhosis [88]. Fetuin-A also disrupts inflammatory pathways and insulin signaling, stimulates the expression of pro-inflammatory cytokines from adipocytes and macrophages, and acts as a novel protein capable of binding free fatty acids to TLR-4 [91].

Given these multiple roles, fetuin-A shows potential as a non-invasive serum biomarker for tracking changes in liver fat and metabolic homeostasis in response to pharmacological treatment of MASLD by monitoring its levels as a reflection of physiological responses.

## 17. Therapeutic Strategies That Influence Fetuin-A Levels

NAFLD/MASLD is widely recognized as the hepatic manifestation of the metabolic syndrome, which is strongly associated with visceral obesity and insulin resistance [93]. Currently, no FDA pharmacological therapies are approved specifically for NAFLD/MASLD or MASH [7]. The only treatment that has been shown to improve fibrosis, steatosis, and hepatic cytolysis in NAFLD/MASLD is weight loss of ≥7–10% of body weight [100], which also improves blood pressure, glycemic control, and lipid profiles, ultimately reducing the risk of cardiovascular disease [7]. Given the strong correlation between body weight, hepatic steatosis, T2D, and elevated fetuin-A levels in both human and animal models, weight loss strategies in NAFLD/MASLD patients with obesity are expected to result in decreased fetuin-A levels, contributing to normalization of hepatic steatosis. An important non-pharmacological approach is a healthy lifestyle that includes physical activity. For example, 3 months of moderate physical activity reduced both circulating and SAT fetuin-A levels in people with obesity or T2D. This reduction is associated with reductions in insulin and adiposity indices in the obese group and glycated hemoglobin in the T2D group [101].

Regarding strategies for the treatment of morbid obesity, bariatric surgery is currently the only effective method for weight loss in this type of patient. In this context, given the association between fetuin-A and obesity and insulin resistance, the effect of gastric bypass on fetuin-A levels has been studied, showing a reduction in the levels of this hepatokine after 16 months, which correlates with the variation of insulin and the HOMA insulin resistance index in obese patients [102]. Another study showed a decrease in fetuin-A levels after Roux-en-Y gastric bypass, mini-gastric bypass, and sleeve gastrectomy in obese patients with and without T2D [103].

In terms of pharmacotherapy, metformin has been recognized as a first-line treatment for T2D due to its ability to suppress hepatic gluconeogenesis by activating adenosine monophosphate-activated protein kinase (AMPK), but it has also been considered as a promising therapeutic agent to reduce hepatic steatosis in MASLD. For example, a study showed that metformin combined with 1,2,3,4,6-penta-O-galloyl-β-d-glucopyranoside (PGG), an inducer of glycine N-methyltransferase (GNMT) expression, which is the most downregulated protein in steatosis liver, reversed all biochemical and histopathological features of MASLD [104]. Another study has described a negative correlation between GNMT and the expression of Angptl8, a hepatokine overexpressed in fatty liver and recognized as a lipogenic factor able to influence triglyceride levels. Indeed, it has been observed that the downregulation of GNMT in hepatocytes of GNMT KO mice resulted in an increase in Angptl8 expression and triglyceride levels, together with an increase in insulin-induced Akt phosphorylation [105]. This may lead to the hypothesis that the beneficial mechanism of metformin combined with PGG observed in MASLD may be related to the inhibition of Angptl8 expression by GNMT. This may suggest that modulation of hepatokines may be considered as a potential therapeutic strategy to treat MASLD. With regard to fetuin-A, many clinical trials have been conducted to evaluate the therapeutic effect of metformin in NAFLD and have shown that it reduces serum fetuin-A levels in patients with NAFLD/MASLD, although this reduction was not associated with corresponding histologic improvements in the liver [87]. However, another study found that metformin was not effective in improving fetuin-A levels in patients with newly diagnosed T2D [106].

On the contrary, both pioglitazone and the glucagon-like receptor agonist (GLP1-Ra) liraglutide have been shown to reduce fetuin-A levels [28,93,106].

## 18. GLP-Ras in the Treatment of MASLD: Modulation of FGF21 and Fetuin-A

GLP-1 RAs are approved drugs for the treatment of obesity in patients with or without T2DM due to their efficacy in promoting weight loss, improving glycemic control, and addressing other cardiometabolic risk factors [100]. Although not approved by the Food and Drug Administration for the treatment of MASLD, these drugs are available for the treatment of obesity and T2D and have been shown to reverse steatohepatitis and to be safe across the spectrum of MASLD with or without fibrosis, reducing the associated cardiovascular risk [107]. These agents have also shown promise in the treatment of patients with T2D and/or obesity and NAFLD/MASLD [7], given their ability to improve blood pressure and lipid profiles, reduce appetite and body weight, and provide a sense of satiety [107].

Among this class of drugs, liraglutide has shown significant improvements in glucose and lipid metabolism, as well as increased insulin sensitivity in both liver and adipose tissue [7]. Clinical trials have shown that treatment with liraglutide can lower plasma aminotransferase levels, reduce liver fat content, and improve steatosis and fibrosis [108,109]. In addition, liraglutide has shown benefit in patients with biopsy-proven MASH [110], highlighting its potential as a therapeutic option for the treatment of NAFLD/MASLD, particularly in patients with T2D and obesity.

Although the exact mechanisms of action of GLP1-Ras remain to be elucidated, it is suggested that in addition to its direct effects on improving glucose and lipid metabolism and reducing steatosis, the hepatic effects of GLP1-Ras may also be mediated by indirect pathways [20] related to weight loss as well as other weight-independent mechanisms. These mechanisms include improvement of hepatocyte insulin resistance and mitocondrial function, adipose tissue lipotoxicity, and adjustment of blood insulin and glucagon concentrations with beneficial effects on lipid and glucose levels [111].

In particular, the effects of GLP1-Ras on lipid metabolism involve their interaction with GLP1-R receptors. Activation of hypothalamic GLP-1Rs in central nervous system neurons leads to a reduction in food intake and subsequent weight loss [112]. Preclinical and clinical studies with GLP1-Ras have correlated improvements in liver steatosis and inflammation with weight loss. For example, treatment with 0.4 mg/kg/day liraglutide for 12 weeks in an obese mouse model of MASH showed improvement in liver histology scores (steatosis, lobular inflammation, hepatocyte ballooning) from baseline along with significant weight loss [113]. A study in patients with biopsy-confirmed MASH and liver fibrosis showed that 72 weeks of treatment with semaglutide, 0.1, 0.2, or 0.4 mg daily, improved MASH in association with dose-dependent weight loss [114]. In the LEAN (Safety and Efficacy in Patients with Non-Alcoholic Steatohepatitis) study, liraglutide 1.8 mg daily was also shown to be safe and to result in histologic resolution of non-alcoholic steatohepatitis after 48 weeks of treatment [110].

Regarding the localization of the GLP-1R in the liver, some studies suggest the absence of GLP-1 receptors in hepatocytes [115,116], while others indicate their presence [117,118,119,120]. However, the lack of specificity of many commercially available reagents or the inability to simultaneously detect the presence of full-length GLP1R/Glp1r mRNA transcripts or canonical GLP-1R signaling responses in hepatocytes makes the available data inconclusive and difficult to interpret or reproduce [112]. Nevertheless, weight loss, together with increased insulin secretion and decreased glucagon expression, mediated the ability of GLP-1Rs to improve hepatic parameters by reducing gluconeogenesis, de novo lipogenesis, triglyceride synthesis, and intrahepatic accumulation, as well as increasing glucose uptake [107,112]. Indeed, the hepatic insulin resistance states in MASLD cause an impairment of hepatic insulin signaling, redirecting glucose from glycogen synthesis to lipogenesis and exacerbating hepatic steatosis. The increased insulin secretion associated with the activation of GLP1-Rs in pancreatic β-cells increases nutrient uptake in adipose tissue rather than in the liver and attenuates adipose tissue lipolysis, thereby reducing the flux of free fatty acids to the liver and the substrate available for de novo lipogenesis. Similarly, although enterocytes do not express GLP1-R, it has been suggested that GLP1-Ras clinically reduces the synthesis and secretion of chilomicron [112].

The potential benefits of these drugs in reversing MASH in patients with T2D, pre-diabetes, and/or obesity may be due to the reduction in steatosis, inflammation, and liver fibrosis by improving ER stress responses, mitochondrial function, and fatty acid oxidation, and by decreasing macrophage recruitment to hepatocytes, common mechanisms involved in liver injury [20,100]. However, behind their metabolic effects, GLP-1 RAs appear to affect hepatic pathways through an indirect mechanism that is not fully understood. One of their indirect pharmacological mechanisms, such as modulation of adipokines and hepatokines, could also be hypothesized behind the critical role of body weight reduction and improvement of glycemic control.

For example, they have been shown to reduce adipose tissue, insulin resistance, and inflammatory markers while increasing the anti-inflammatory adipokine adiponectin [107]. Consistent with these findings, the reduction in hepatic steatosis and endoplasmic reticulum oxidative stress in liraglutide-treated mice appeared to occur independently of body weight loss [100]. In this context, given the involvement of PPAR-gamma and adiponectin in the improvement of insulin resistance [121], it is possible to hypothesize that the upregulation of adiponectin expression observed after treatment with GLP-1 administration, especially liraglutide [100,122], and exenatide [123], may mediate their ability to improve insulin resistance and thereby reduce hepatic steatosis. For example, the beneficial effects of PPAR-γ agonists such as pioglitazone in prediabetic and diabetic patients with MASH confirmed by liver biopsy, despite weight gain, seem to be determined by the concomitant increase in adiponectin levels and redistribution of fat from visceral to subcutaneous [124].

An upregulation of hepatic FGF21 production stimulated by GLP1-Ras exenatide or liraglutide was also observed in plasma and liver tissue of diabetic mice and in serum of T2D patients. This finding has been proposed as a novel extrahepatic mechanism mediating the glucose-lowering effect of GLP1-Ras through inhibition of gluconeogenesis and suggests the presence of FGF21 resistance compensation [125]. Similarly, in a mouse model of obesity and diabetes, systemic administration of liraglutide or exenatide increased hepatic FGF21 expression and contributed to the amelioration of obesity and hyperglycemia [125,126]. In contrast, 12 weeks of liraglutide administration decreased circulating levels of FGF21, in addition to reducing body weight and liver fat. These results were observed in patients with both T2D and NAFLD/MASLD, in which the state of FGF21 resistance or insensitivity is higher, caused by the abnormal production of this hepatokine by “fatty hepatocytes” [24,127].

The different levels before and after treatment may be related to the degree of resistance to this hepatokine, which may partially explain the difference in results. For example, the reduction in its level may suggest attenuating FGF21 liver resistance, as well as the upregulation of FGF21 production will lead to an improved lipid profile, attenuating FGF21 resistance. In both cases, these findings may suggest that GLP1-Ras ameliorates the resistance and action of FGF21, restoring its physiological levels and activity [127]. In fact, FGF21 could correct multiple metabolic abnormalities both in vitro and in vivo as part of a strategy to improve hepatic steatosis and inflammation [24].

Regarding the investigation of modulation of fetuin-A levels in the context of metabolic diseases, both the antidiabetic drugs liraglutide and pioglitazone were found to reduce fetuin-A levels [28]. In a study of patients with T2D and NAFLD/MASLD, liraglutide therapy was associated with reductions in both liver fat content and fetuin-A levels compared with pioglitazone. As there is evidence that 12 weeks of aerobic exercise significantly reduced serum fetuin-A levels in T2D patients, this reduction was mainly attributed to weight loss. However, elevated fetuin-A levels were associated with body weight, waist circumference, and ectopic liver fat accumulation, while its reduction also correlated with decreased liver fat content [93]. This finding confirms its role as a marker of liver fat accumulation and overall metabolic health and suggests that lipid overload in MASLD may influence fetuin-A expression.

Considering the positive association between fetuin-A levels and liver fat content and the observation that free fatty acid seems to enhance fetuin-A expression in hepatocytes [56], it is important to first consider the involvement of hepatic steatosis. In this context, the improvement of hepatic steatosis after GLP1-Ras treatment can be considered as a possible indirect mechanism to explain the reduction in circulating fetuin-A levels [100].

However, an increase in adiponectin levels has also been observed after GLP1-Ras treatment [128], as well as after treatment with the dual agonist GIP-GLP-1 tirzepatide [129,130]. In particular, it has been reported the upregulation of adiponectin secretion in culture adipocytes by exendin-4, suggesting its possible role in enhancing the level of adiponectin mRNA both through the protein kinase a (PKA) and both with the reduction in pro-inflammatory adipokines expression IL-6 and MCP-1 [129]. PKA activators have been recognized as the signaling pathway required for PPARy activation, which is responsible for adipocyte differentiation and adiponectin secretion. These may explain why the ability of GLP1-Ras to enhance PPARy may be an indirect mechanism mediating their insulin-sensitizing and anti-inflammatory effects [131,132,133].

Consistent with this hypothesis, liraglutide has been implicated in the stimulation of pre-adipocyte differentiation into mature adipocytes and adipogenesis. In fact, since suppression of adipogenesis, associated with an increase in adipocyte size, is associated with increased insulin resistance, upregulation of adipocyte differentiation results in increased glucose uptake and secretion of adiponectin, improving systemic insulin sensitivity and reducing ectopic lipid accumulation in the liver, heart, or muscle and weight gain [134].

For example, a study that observed upregulation of adiponectin after treatment with pioglitazone also hypothesized that stimulation of PPARy in adipocytes leads to upregulation of adiponectin as an indirect mechanism of fetuin-A suppression. Indeed, the anti-diabetic, anti-atherosclerotic, and anti-inflammatory properties of adiponectin reduce lipid storage in the liver and muscle, improve insulin sensitivity, and alleviate fatty liver disease by suppressing fetuin-A levels [135]. However, the existence of a regulatory mechanism whereby adiponectin exerts an inhibitory effect on fetuin-A expression through the AMPK pathway has also been reported [37,136,137]. Consistent with this, the decreased serum fetuin-A levels observed in T2D patients after 12 weeks of aerobic exercise training were also associated with increased adiponectin levels [138].

FGF21 analogs also appear to stimulate adiponectin secretion from adipose tissue [86,139]. Therefore, the investigation of the novel GLP-1/FGF21 axis and the possible modulation of adiponectin and fetuin-A levels could be considered as a potential mechanism to explain the beneficial effects of GLP1-Ras in the pathophysiology of MASLD.

## 19. FGF21/GLP-1 Axis in the MASLD: Fetuin-A Can Be the Target of the Dual Agonist?

The heterogeneous and slow natural history of liver disease requires long-term studies to demonstrate the effect of an intervention on clinical outcomes, and studies are currently limited. As the management of cardiometabolic comorbidities plays a critical role in influencing disease progression, appropriate management of all these comorbidities may be critical [140].

The biological effects of GLP-1 in suppressing lipogenesis, reducing hepatic steatosis and pro-inflammatory responses, and improving cardiovascular outcomes, together with the recognition of FGF21 as a key regulator of metabolism that can improve lipid homeostasis and insulin sensitivity and has an antifibrotic effect, make it an attractive candidate for the treatment of MASLD. In fact, as previously reported, there appears to be a synergistic interplay between these agonists, with FGF21 appearing to mediate the effects of GLP-1R in inhibiting hepatic glucose output, reducing liver fat, and decreasing inflammation, while hepatic production of FGF21 could be upregulated by GLP-1R agonists such as exenatide and liraglutide. Recently, there has been growing interest in developing a drug that can target both GLP-1 and FGF21. This may represent an ideal therapeutic strategy for MASLD, taking advantage of their complementary mechanisms of action to reduce liver inflammation and hepatocyte injury, improve underlying insulin resistance, and exert anti-fibrotic effects [141] (Figure 4).

The effect of a low-dose combination of a GLP-1 analog and FGF21 was studied in a mouse model followed by an atherogenic diet and showed a reduction in atheromatous plaque area in the aorta, decreased body weight and blood glucose levels, and upregulation of the mitochondrial protein UCP1. This suggests that this treatment reduces obesity, improves insulin sensitivity, and attenuates the progression of atherosclerosis by improving inflammation. In addition, an increase in adiponectin and a decrease in leptin levels have been suggested as underlying mechanisms for the anti-inflammatory and anti-atherosclerotic activities [142].

The same results, showing promising effects in preclinical studies, have been obtained with the long-acting dual agonists against FGFR and GLP-1R, such as YH25724, GLP-1-Fc-FGF21 (made with an immunoglobulin Fc-fused protein), GLP-1-ELP-FGF21 (fused with an elastin-like polypeptide linker), and ZT003 (made by fusion of GLP-1, anti-HSA nanobody, and FGF21) [143,144,145]. In a diet-induced MASH model, YH25724 reduced body weight, liver enzymes, plasma, and hepatic triglycerides and decreased liver fibrosis and inflammation, confirming anti-steatotic, anti-inflammatory, and anti-fibrotic effects [141]. The GLP-1-Fc-FGF21 dual agonist, which enhances β-klotho binding property, has also been shown to induce a greater hypoglycemic effect with a more significant reduction in body weight and food intake than single agonists, suggesting a synergistic action of FGF21 and GLP-1 receptors at the central nervous system level to suppress appetite and reduce caloric intake, improve insulin sensitivity, and show a thermogenic effect. In addition, more significant therapeutic effects have been observed in attenuating the progression of MASH in terms of liver function, lipid profiles, and overall NAS scores, which include steatosis, lobular inflammation, and hepatocyte ballooning [144,146].

Given that native FGF21 acts on adipose tissue not only to increase glucose uptake and regulate lipid metabolism but most importantly to stimulate adiponectin secretion [139], the study of FGF21 analogs as a treatment MASLD and its more advanced form MASH has shown to increase adiponectin secretion from adipose tissue [86,147,148]. This may be a critical mechanism mediating the metabolic, anti-inflammatory, anti-diabetic, and antifibrotic effects of dual GLP-1-FGF21 agonists.

Given the critical role of fetuin-A in insulin resistance, obesity, T2D, and MASLD, and the beneficial effects of adiponectin on fatty acid oxidation, lipid reduction, and improvement of insulin resistance through activation of 5′-AMP-activated protein kinase (pAMPK), it is not surprising that adiponectin and fetuin-A serum concentrations are inversely correlated in these diseases.

A correlation between the reduction in adiponectin levels in diabetic obese mice and the higher levels of fetuin-A has been reported, whereas an increase in adiponectin levels has been observed after knockdown of the fetuin-A gene. In addition, fetuin-A overexpression has been shown to inhibit PPARy expression through upregulation of the Wnt3a pathway, which affects adiponectin secretion, and its insulin-sensitizing role mediated by AMPK activation. Taken together, these results may implicate fetuin-A in the reduction in adiponectin levels observed in metabolic disorders, including MASLD [92,149].

Since it has been suggested that the novel GLP-1/FGF21 axis plays a critical role in mediating the beneficial effects of GLP-1-based pharmacotherapy in people with obesity [150], these overall findings allow the hypothesis that FGF21 likely suppresses fetuin-A levels directly by acting on adipocytes through the upregulation of adiponectin levels mediated by PPARy activation.

The novel GLP-1/glucose-dependent insulinotropic polypeptide (GIP) receptor agonist tirzepatide has also demonstrated superior metabolic efficacy, including improvements in serum aminotransferase levels and liver fibrosis scores, in patients with T2DM and/or obesity [151]. These findings suggest that similar to other GLP1-Ras, tirzepatide may affect fetuin-A levels as a mechanism of action.

Taken together, these findings may provide a likely explanation for the beneficial effects of GLP1-Ras observed in the overall metabolic abnormalities associated with MASLD and involved in its progression to MASH through the GLP1-FGF21-fetuin-A axis.

## 20. Conclusions

MASLD, the most common liver disease, often precedes other metabolic conditions such as obesity, insulin resistance, and T2D. With the recent focus on abnormal circulating levels of adipokines as markers of risk for type 2 diabetes and cardiovascular and metabolic disease, there is increasing evidence for altered protein secretion patterns in the liver of MASLD, analogous to dysregulated adipose tissue.

The recognition of the liver as an endocrine organ capable of secreting hepatokines that are both markers of the disease and involved in its pathophysiology, particularly insulin resistance and subclinical inflammation, has opened new avenues for understanding and potentially treating the metabolic disorders associated with MASLD.

For researchers, this review provides new information on how the crosstalk between adipokines such as adiponectin and hepatokines such as fetuin-A and FGF21 may be independently involved in the pathogenesis of MASLD. Beyond the known altered levels of the protective adipokine adiponectin in metabolic diseases, abnormal secretion of hepatokines such as FGF21 and fetuin-A in response to metabolic stress has been shown to disrupt insulin signaling, glucose and lipid metabolism, inflammation, and lipotoxicity, contributing to the development and progression of MASLD. These hepatokines are emerging as promising predictive biomarkers for both MASLD and its progression to MASH and associated metabolic complications.

Modulation of these hepatokines could help restore energy homeostasis, improve insulin sensitivity, and reduce inflammation and resistance in target tissues.

This review proposes the hypothesis that their modulation may mediate the effects of GLP1-Ras on adipose tissue, inflammation, glucose and lipid metabolism, and cardiovascular disease, providing novel endogenous mechanisms beyond the classical actions.

Further research, particularly human studies, is needed to validate these effects and to explore the mechanisms by which GLP1-FGF21 dual agonist pharmacotherapy may modulate fetuin-A through adiponectin to contribute to the management of metabolic and liver diseases. However, this provides potential insight into the mechanisms by which this type of pharmacotherapy may manage metabolic and liver-related conditions such as MASLD and may emerge as effective therapeutic strategies to restore metabolic balance and liver function.

## Figures and Tables

**Figure 1 ijms-25-10795-f001:**
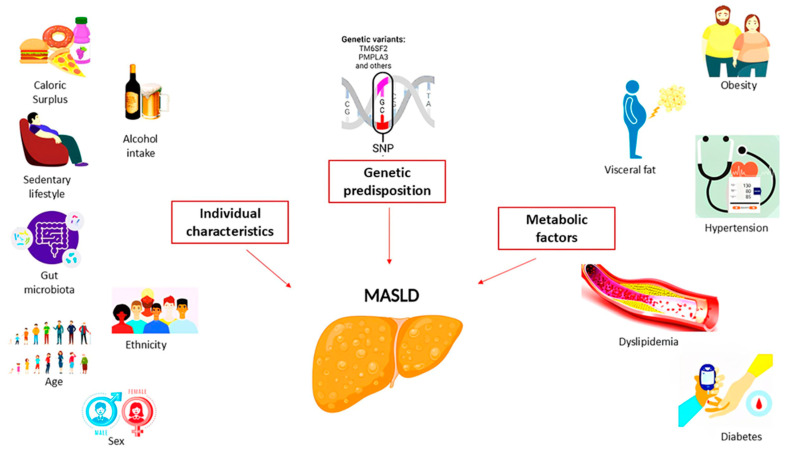
Pathogenetic mechanisms involved in the development of MASLD.

**Figure 2 ijms-25-10795-f002:**
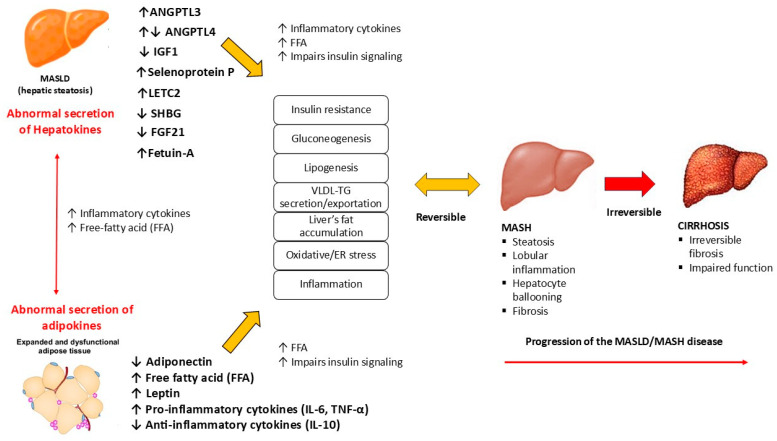
The role of hepatokines in the MASLD metabolic dysfunction. MASLD: metabolic dysfunction-associated steatotic liver disease; MASH: metabolic dysfunction-associated steatohepatitis; FFA: free-fatty acid; VLDL: very low-density lipoprotein; TG: triglycerides; TNF-α: tumor necrosis factor α; IL-10: interleukin-10; IL-6: interleukin 6.

**Figure 3 ijms-25-10795-f003:**
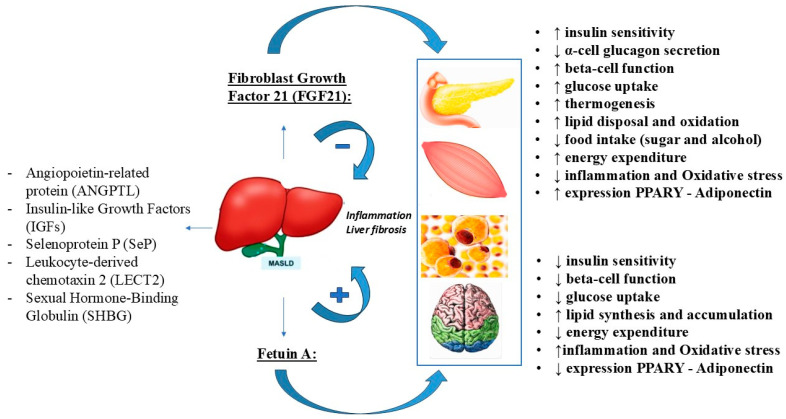
The opposite role of FGF21 and fetuin-A in the MASLD.

**Figure 4 ijms-25-10795-f004:**
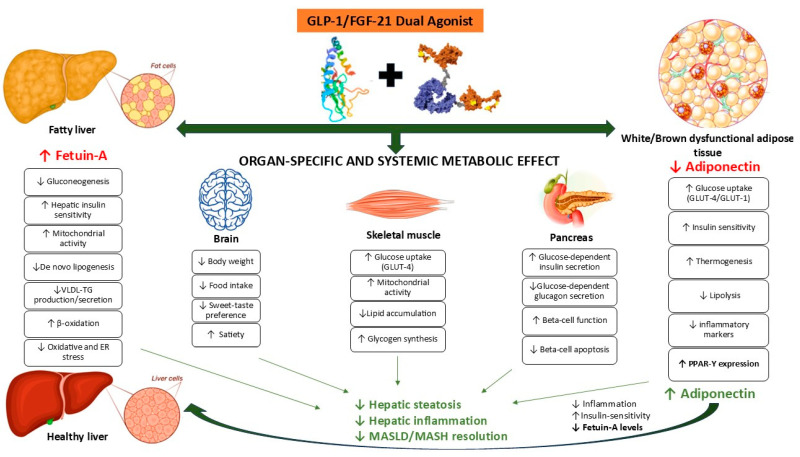
The GLP1-FGF21 dual agonist in the MASLD and the target of fetuin-A.

**Table 1 ijms-25-10795-t001:** The role of hepatokines in the MASLD metabolic dysfunction.

	Expression	Change with Steatosis	Metabolic Functions	Reference
ANGPTL3	Liver	Increased	Inhibits LPL activity, increases plasma triglycerides TG and FFA, and increases TG-VLDL uptake. Decreases glucose uptake and promotes insulin resistance. Increases cardiovascular risk.	[6,28,37,40,41]
ANGPTL4	Liver Skeletal muscle Heart	Debated	Inhibits LPL activity, increases lipolysis, downregulates blood TG clearance, and causes hepatic steatosis. Role in regulation of glucose metabolism is unclear.	[23,37,42,43]
ANGPTL6	Liver	Increased with impairment signaling (resistance)	Increases insulin sensitivity and energy expenditure (muscle). Decreases gluconeogenesis (liver). Increases AMPK.	[6,34,37]
IGF1	Several tissues (especially liver)	Decreased	Improves insulin sensitivity (muscle).	[6,37]
SeP	Liver	Increased	Interferes with insulin signaling and glucose metabolism, increases insulin resistance, impairs β-cell function, and decreases glucose uptake (muscle). Causes carotid intima media thickness.	[6,23,34,37,44,45]
LECT2	Liver (mainly) WAT, neurons, and white blood cells	Increased	Interferes with insulin signaling. Promotes lipid accumulation (liver). Increases liver inflammation and fibrosis (muscle).	[6,28,46,47]
SHBG	Liver	Decreased	ER protection. Downregulate lipogenesis Reduced hepatic steatosis, inflammation	[31,44,48]
FGF21	Liver (mainly) Intestine, heart, kidney, pancreas, WAT, and BAT	Increased	Promotes glucose uptake (WAT and muscle). Stimulates thermogenesis (BAT). Increases FFA oxidation and insulin sensitivity (liver, muscle). Decreases steatosis, VLDL uptake, and lipogenesis (liver). Decreases alcohol and sugar intake. Increases energy expenditure and decreases body weight (CNS). Increases insulin secretion (beta cells).	[24,49,50,51,52,53,54,55]
Fetuin-A	Liver (mainly) WAT	Increased	Promotes inflammation (monocytes and adipocytes). Inhibits adiponectin production. Interferes with insulin receptor phosphorylation (liver, WAT, and skeletal muscle) and causes insulin resistance.	[28,37,56,57,58]

ANGPTL3: angiopoietin-like protein; ANGPTL4: angiopoietin-like protein 4; ANGPTL6: angiopoietin-like protein 6; IGF-1: insulin-like growth factor 1; SeP: selenoprotein P; LECT2: leukocyte cell-derived chemotaxin 2; SHBG: sex hormone-binding globulin; FGF21: fibroblast growth factor 21; LPL: lipoprotein lipase; TG: triglycerides; FFA: free-fatty acid; VLDL: very low-density lipoprotein; AMPK: adenosine monophosphate-activated protein kinase; MASLD: metabolic dysfunction-associated steatotic liver disease; WAT: white adipose tissue; BAT: brown adipose tissue; CNS: central nervous system.

## Data Availability

The original contributions presented in this study are included within the article; further inquiries can be directed to the corresponding author.

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
