# Peer review of "Hepatokines and MASLD: The GLP1-Ras-FGF21-Fetuin-A Crosstalk as a Therapeutic Target"

_ijms, 2024, doi:10.3390/ijms251910795_

Round 1

Reviewer 1 Report

Comments and Suggestions for Authors

The current manuscript summarized the roles of "hepatokines" such as FGF21 and Fetuin-A in MASLD. It is an important topic for MASLD, since it is an important field for drug development. However, it is very strange that why the authors choose these hepatokines to report, since there are several hepatokines. I suggest the authors change the tile with “The roles of the hepatokines in the metabolic dysfunction of MASLD”, and they can add more details about other hepatokines, then focus on FGF21 and Fetuin-A. Besides, recent publications (BMC Gastroenterol. 2024 Jul 18;24(1):226; United European Gastroenterol J. 2024 Jun 18. doi: 10.1002/ueg2.12534) should be included.

Comments on the Quality of English Language

NA

Author Response

Reviewer 1

The current manuscript summarized the roles of "hepatokines" such as FGF21 and Fetuin-A in MASLD. It is an important topic for MASLD, since it is an important field for drug development. However, it is very strange that why the authors choose these hepatokines to report, since there are several hepatokines. I suggest the authors change the tile with “The roles of the hepatokines in the metabolic dysfunction of MASLD”, and they can add more details about other hepatokines, then focus on FGF21 and Fetuin-A. Besides, recent publications (BMC Gastroenterol. 2024 Jul 18;24(1):226; United European Gastroenterol J. 2024 Jun 18. doi: 10.1002/ueg2.12534) should be included.

Response: We sincerely thank the reviewer for taking the time to review this paper and for providing constructive comments and feedback that helped us improve the paper.

Regarding the question of why we chose to report these hepatokines (FGF21 and Fetuin-A) rather than several others, we strongly agree with the reviewer's observation and thank him for bringing the importance of this issue to our attention. We have therefore accepted his kind suggestion to expand our analysis by adding paragraphs describing the other hepatokines from lines 364-473.

In addition, we have added a description in lines 313-316 and changed the sentence as reported in lines 340-359.

As FGF21 and fetuin-A are reported to be the most studied hepatokines in humans and animals, this observation also led us to deepen the description of their hypothalamic modulation by GLP1-Ras as a treatment for MASLD. For this reason, we have added two paragraphs from lines 868-1014 and from 1016-1091 in which we have discussed this topic. We strongly agree with the reviewer's suggestion to change the title, and decided to modify it as follows: “Hepatokines and MASLD: the GLP1-Ras-FGF21-Fetuin A cross-talk as a therapeutic target”, in order to include this cross-talk as focus.

We also apologize for not including the recent publications "BMC Gastroenterol. 2024 Jul 18;24(1):226" as suggested by the author, and we have added it in lines 786-805. Regarding the cited article "United European Gastroenterol J. 2024 Jun 18. doi: 10.1002/ueg2.12534", we have decided to add its results in other parts of the paper, e.e.g. in lines 954-957, 963-965, as well as in lines 254-260, where we have added a short description of the genomics of fatty hepatocytes reported in this study, emphasizing the importance of this study.

All changes are highlighted in yellow.

Reviewer 2 Report

Comments and Suggestions for Authors

The review article „The role of the hepatokines FGF21 and Fetuin-A in the metabolic dysfunction of MASLD“ by Ilaria Milani et al., provides a valuable overview of the role of hepatokines FGF21 and Fetuin-A in the metabolic dysfunction of MASLD. While the article is well-written, it doesn't offer any new insights compared to existing literature on the topic. Moreover, the manuscript over-relies on the findings of cited papers without providing a critical discussion. For these reasons, the manuscript needs to be significantly improved before publication.

Majors:

1)    The extensive discussion on terminology detracts from the primary focus of the review. Moreover, this shift has been extensively discussed in the past years.

2)    The manuscript would benefit significantly from the inclusion of additional figures and tables. Figures that illustrate mechanisms, pathways, and interactions between hepatokines and MASLD could provide valuable visual summaries. Also, tables that summarize key data or compare findings across studies would enhance the readability and impact of the review.

3)    Including comprehensive data analysis in the manuscript and incorporating analyses from publicly available databases, transcriptomic, or proteomic data could reveal the underlying mechanisms of MASLD and provide a more robust discussion.

4)    The section on GLP-1 receptor agonists would benefit from a more detailed explanation of the indirect mechanisms by which these drugs might affect fetuin-A levels and MASLD.

5)    Addressing the limitations of current studies on fetuin-A, such as variability in results or potential confounding factors, would provide a more balanced view and highlight areas where caution is needed in interpreting the data.

6) The manuscript lacks a clear hypothesis or research focus, what is new and what this review adds to the knowledge. While the role of hepatokines such as FGF21 and fetuin-A in MASLD is discussed (where we find many reviews in PubMed), the specific research question or hypothesis underlying the study is neither explicitly stated nor clearly presented.

Minors:

1)    On page 3 the first paragraph differs from the original layout.  

Comments on the Quality of English Language

Author Response

Reviewer 2

The review article „The role of the hepatokines FGF21 and Fetuin-A in the metabolic dysfunction of MASLD“ by Ilaria Milani et al., provides a valuable overview of the role of hepatokines FGF21 and Fetuin-A in the metabolic dysfunction of MASLD. While the article is well-written, it doesn't offer any new insights compared to existing literature on the topic. Moreover, the manuscript over-relies on the findings of cited papers without providing a critical discussion. For these reasons, the manuscript needs to be significantly improved before publication.

Response

We sincerely thank the reviewer for his valuable time and effort in reviewing our manuscript. We have revised the manuscript according to the thoughtful and constructive comments and suggestions, which helped us to clarify the focus of this review, hoping to having improve its quality and novelty. In this response, we provide a detailed point-by-point response to the reviewer's suggestion, and all changes have been highlighted in yellow in the article.

Majors:

1)The extensive discussion on terminology detracts from the primary focus of the review. Moreover, this shift has been extensively discussed in the past years.

Response: We strongly agree with the reviewer's comment and note that the introduction discusses a lot of the MASLD terminology issue, which is missing from the central focus of the review. For this reason, we have decided to delete the parts related to the terminology discussion and replace them with a brief description of the focus of the review in new lines 52-65, highlighted in yellow.

2)    The manuscript would benefit significantly from the inclusion of additional figures and tables. Figures that illustrate mechanisms, pathways, and interactions between hepatokines and MASLD could provide valuable visual summaries. Also, tables that summarize key data or compare findings across studies would enhance the readability and impact of the review.

Response: We thank the reviewer for this valuable suggestion and have added two figures (Figure 2 line 232, Figure 4 line 1035) and one table (Table 1 line 343) as suggested. We have also added more information about the mechanisms in the text, from lines 150-167, 174-178.

3)    Including comprehensive data analysis in the manuscript and incorporating analyses from publicly available databases, transcriptomic, or proteomic data could reveal the underlying mechanisms of MASLD and provide a more robust discussion.

Response: We thank the reviewer for this careful and valuable review. We have added a new paragraph "The analysis of metabolic organ-secreted factors in MASLD " in which we have tried to discuss the altered gene and protein expression reported by proteomic and transcriptomic data analysis in lines 247-305.

4)    The section on GLP-1 receptor agonists would benefit from a more detailed explanation of the indirect mechanisms by which these drugs might affect fetuin-A levels and MASLD.

Response: We strongly agree with the reviewer for this careful consideration, and we apologize for missing it. We have added a hypothetical explanation of the indirect mechanisms by which GLP1-Ras appears to affect fetuin-A levels in lines 966-1009.

5)    Addressing the limitations of current studies on fetuin-A, such as variability in results or potential confounding factors, would provide a more balanced view and highlight areas where caution is needed in interpreting the data.

Response: We thank the reviewer for this valuable suggestion. We have added a part in the text which can provide careful attention about the biphasical role of fetuin-A from lines 741-758 and we have added limitations of some studies reported, such as in lines 734-735, 774-776, 778-781, 795-796, 801-805.

6) The manuscript lacks a clear hypothesis or research focus, what is new and what this review adds to the knowledge. While the role of hepatokines such as FGF21 and fetuin-A in MASLD is discussed (where we find many reviews in PubMed), the specific research question or hypothesis underlying the study is neither explicitly stated nor clearly presented.

Response: We thank the reviewer for this thoughtful comment, which raises an important issue and prompted us to clarify the focus of this review. We have modified the title as follows: “Hepatokines and MASLD: the GLP1-Ras-FGF21-Fetuin A cross-talk as a therapeutic target” and we have modified the paragraph on the modulation of fetuin-A (lines 848-860, 863-866). We have also added two new paragraphs. One entitled "GLP1-Ras in the treatment of MASLD: modulation of FGF21 and Fetuin-A" from lines 868-1014, and another entitled "FGF21/GLP-1 axis in MASLD: Fetuin-A can be the target of the dual agonist?" from lines 1016-1091. Given the recent consideration of GLP1-Ras as a possible treatment for MASLD, we have tried to add novelty to this review by hyphenating how the GLP1-Ras can have beneficial effects through the modulation of FGF21 and Fetuin-A. Thus, FGF21 can mediate the effects of GLP1, and for this reason, the dual agonist FGF21/GLP-1 can have a sinergistic effects on the modulation of fetuin-A, but also adiponectin levels. This cross-talk may be considered as a possible target therapy for MASLD in the future, but further research is needed.

 We have also changed the abstract from lines 25-35, and the conclusion of the review from lines 1097-1099, 1102-1103, 1105-1107, 1116-1125 to emphasize the focus.

Minors:

  • On page 3 the first paragraph differs from the original layout.  

Response: We thank the reviewer for this careful observation. We have tried to change it and use the same layout for all text, as kindly suggested.

Round 2

Reviewer 1 Report

Comments and Suggestions for Authors

NO further comments.

Author Response

We would like to thank the Reviewer for the valuable feedback that helped us to improve the quality of our manuscript.
